# Technical note: Graph theory-based heuristics to aid in the implementation of optimized drinking water network sectorization

Marius Møller Rokstad[1], Karel van Laarhoven [2]

[1]Department of Civil and Environmental Engineering, Norwegian University of Science and Technology (NTNU), Trondheim, 7491, Norway
[2] KWR Water Research Institute, Nieuwegein, 3433, The Netherlands

*Correspondence to*: Marius Møller Rokstad (marius.rokstad@ntnu.no)

**Abstract.** Drinking water distribution networks form an essential part of modern-day critical infrastructure. Sectorizing a network into district metered areas is a key technique for pressure management and water loss reduction. Sectorizing an existing network from scratch is, however, an exceedingly complex design task that designs in a well-studied general mathematical problem. Numerical optimization techniques such as evolutionary algorithms can be used to search for near-optimal solutions to such problems, but doing so within a reasonable timeframe remains an ongoing challenge. In this work, we introduce two heuristic tricks that use information of the network structure and information of the operational requirements of the drinking water distribution network to modify the basic evolutionary algorithm used to solve the general problem. These techniques not only reduce the time required to find good solutions, but also ensure that these solutions better match the requirements of drinking water practice. Both techniques were demonstrated by applying them in the sectorization of the actual distribution network of a large city.

## 1 Introduction

Drinking water distribution systems (DWDS) form an essential part of modern-day critical infrastructure. As these large pipe networks are often hard to reach and interwoven with other urban infrastructure, modifying them is an arduous and expensive task. Regardless, drinking water utilities need to consider applying new designs their networks in order to, for instance, improve service, reduce water losses and to prepare in a resilient way for an uncertain future.

Here, we focus on one specific aspect of network redesign: sectorization. Sectorization entails dividing a DWDS into separate subnetworks, either by closing the boundary pipes between them or by outfitting the boundaries with pumps, pressure reduction valves or flow meters. As a result, the water balance of individual subnetworks or District Metered Areas (DMAs) can be obtained and the pressures in the DMAs can be managed separately.

In many countries, sectorization into DMAs has been a key strategy for leak detection and water loss reduction through pressure management (UK Water Authorities Association, 2008; Farley, 2007; Morrison et al. 2007). In other countries, such as Norway and the Netherlands, however, water utilities have only recently started to consider the advantages of

sectorization. Not having integrated sectorization in their network design process the start, these utilities now face the challenge of designing an efficient division of their complete infrastructure at once. This will involve balancing many different criteria, prime among them the costs of placing or removing network components and the reduction in hydraulic performance that follows from decreased connectivity in the network. Moreover, as DWDS typically are huge, meshed

systems, there often are an overwhelming number of different approaches to dividing up the network.

The key challenge of network sectorization lies in finding ways to efficiently divide the network in as many DMAs as possible with as few changes (which are costly) to the network as possible. This essentially is a version of the (np-hard) minimal k-cut problem (Kim et al. 2011). In the past decades, problems such as these have inspired extensive research on the application of numerical optimization techniques to aid in various aspects of DWDS design (Maier et al., 2014; Mala-

Jetmarova et al. 2018). The literature contains a multitude of examples of various methods applied to achieve effective sectorization of water distribution networks (Alvisi 2015; Brentan et al. 2018; Ciaponi et al. 2016; Diao et al. 2013; Diao et al. 2016; Laucelli et al. 2017; Liu and Han; 2018; Hajebi et al. 2016; di Nardo et al. 2016; Vasilic et al. 2020; Zhang et al. 2016; Zhang et al. 2017). Such techniques allow drinking water experts to explore their options in a systematic, automated way and to subsequently substantiate their choice for specific, optimal solutions. One particularly versatile technique that has

received thorough attention in this context is that of Genetic Algorithms (GA) (Holland, 1975; Goldberg, 1989) and other members of the overarching family of evolutionary algorithms (EA).

While EA are known to be suitable for sectorization type problems, one of their limitations is the unreasonable amount of function evaluations required to converge to an optimal solution properly (Kim et al., 2011). Moreover, when it comes to practical application of EA, improving their searching behaviour in terms of computational speed and quality of results

remains an ongoing challenge (Maier et al. 2014, van Thienen et al. 2018). To this end, the various mechanisms of the classic genetic algorithm are commonly expanded, replaced or combined with heuristic tricks or complete heuristic algorithms to improve performance (Maier et al., 2014; Krasnogor and Smith, 2005; El-Mihoub et al. 2006; van Laarhoven et al. 2018). The algorithms which include these are commonly referred to as Hybrid Genetic Algorithms (HGA) or Memetic Algorithms (MA).

In this paper, we report two HGA techniques that have aided in the successful application of EA for sectorizing real-life DWDS of large towns in Norway and the Netherlands. Both techniques first use graph theory algorithms to extract aspects of the network structure in a formal way. This information on network structure is used to guide the searching behaviour of the EA towards structures that are preferable in a DWDS according to the criteria of water utility experts. Making the design criteria of utility experts explicit in this way enhances trust in the technique and thereby enhances the chances of practical

implementation of numerical optimization. Secondly, computational time required to find suitable solutions is reduced, so that practical application becomes feasible.

## 2 Methods

### 2.1 Basics of EA and their application to sectorization

EA are a type of optimization algorithm inspired by concepts from genetics. The general principle behind this type of algorithm is shown schematically by the blue boxes in Figure 1. First, a collection of possible solutions is created (a population of individuals). The solutions are tested for their performance according to the user's performance criteria. The least successful solutions are discarded (natural selection) and the collection is supplemented with new solutions. The new solutions are generated by creating small variations in well-performing solutions (mutation) or by combining elements from two well-performing solutions (reproduction, or crossover). The process is then repeated several times, gradually improving the quality of solutions (evolution).

If individual solutions are judged based on only one performance criterium, selecting the final candidate is a matter of selecting the solution with the highest performance. If multiple performance criteria are used, however, it may occur that these criteria are at odds, so that a choice must be made that accepts a trade-off between the two. This trade-off is typically represented by a Pareto front, a graph that scatters individual solutions on two or more axes that correspond to their scores according to the different criteria.

The white boxes in Figure 1 illustrate a basic way in which an EA can be applied to find solutions to the sectorization problem, i.e. to find ways to divide the network into subnetworks with as few boundaries between them as possible:

- **Initialization**: individual solutions are defined by assigning every demand node in a DWDS to a given DMA.

- **DMA performance criteria**: typical aspects that are important for the performance of a solution are for instance the sizes of the individual DMAs and the number of boundaries between them. To determine these aspects, a specific representation of solutions is needed that includes the graph topology of the network. In other words: the solution must contain not only information on the demand nodes of the DWDS, but also on the pipes between them. A specific representation may also include more detailed information on the functional properties of the DWDS – effectively forming a complete hydraulic model – so that the hydraulic performance of a solution might be assessed (for instance in terms of reduced supply capacity once a certain pipe between DMAs is closed).

- **DMA performance constraints**: rather than being used as performance criteria that drive the direction of optimization, solution properties may also be made subject to constraints or boundary. This ensures that networks keep meeting practical requirements while their configuration is changed to optimize the performance criteria. This could be as simple as putting a minimum or maximum on the value of a performance criterium for a solution to be considered viable. Far more complex constraints may be useful or necessary, however. The topic of chapter 2, for instance, is ensuring that connectivity, redundancy and pressure requirements throughout the network are met when every DMA boundary is outfitted with a pressure reduction valve of a particular setting (a constraint that requires multiple hydraulic simulations to evaluate for a single possible solution).

- **Mechanisms to create candidate variation**: basic mutation can be achieved by splitting DMAs in two or by merging adjacent DMAs. Crossover can be achieved by taking specific DMAs from two solutions to construct a new solution (paying attention to smoothing possible gaps between solutions).
- **Results and Pareto front**: The two criteria used here are mutually exclusive: smaller DMAs are beneficial, but require more boundaries to realize, which is not preferred. As such, the individual results should be ordered in a Pareto front from which a solution must then be chosen based on criteria outside the optimization process.

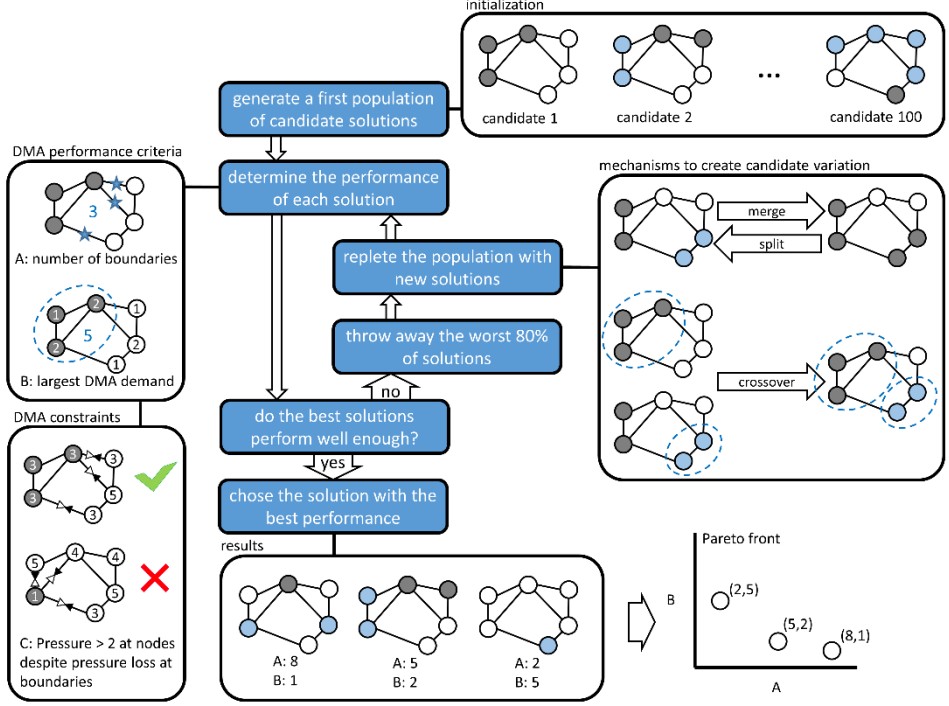

**Figure 1. Schematic representation of the evolutionary optimization algorithm used. The blue boxes summarize the general operation of the algorithm. The white boxes illustrate a basic implementation of the algorithm used to optimize a DMA configuration.**

## 2.2 Shortest independent paths identification algorithm for reliability conservation and search space reduction

One specific reason to sectorize a network may be to divide it into pressure management zones. These are subnetworks, separated by pressure reduction valves, aimed at reducing the pressure throughout the network. In this case, every boundary between zones has an implication for the hydraulic performance of the network. The hydraulics of the network, however, are subject to many practical constraints and, indeed, regulations. It must for instance be ensured that, despite the introduction of pressure reduction valves, the network is able to supply sufficient water not only in the nominal situation, but also during a pipe failure calamity or while a fire hydrant is in use.

Checking whether a particular solution will meet these constraints will require a multitude of hydraulic simulations under different scenarios. The computational time required to verify this could easily be in the order of tens of hours for a single

solution when a realistically large distribution network is considered. As such, the time required to check the performance of individuals is prohibitively large for an EA to be used (in which easily millions of individuals need to be evaluated throughout the search). The need to check whether a particular solution for sectorization of the drinking water network will be in violation of the performance requirements set by the utility or legislation has been limiting Trondheim municipality's capacity to optimize their sectorization with respect to pressure management, as the number of hydraulic simulations and computational time would be impractically high, thus making it virtually impossible for the utility to identify a globally effective solution for pressure management. The objective of the algorithm suggested here is to provide an alternative way of ensuring the hydraulic requirements of solutions while optimizing a sectorization with pressure control zones in mind.

The core of the approach is to evaluate to which extend individual nodes in a water distribution system are served by multiple, independent paths, as this is a measure of reliability and system robustness. This is achieved by representing the network as a bi-directional weighted graph, identify the shortest path between source and demand node using Dijkstra's algorithm, consecutively changing the weights of the graph, and re-running the shortest path algorithm so that the paths which are maximally independent are identified.

### 2.2.1 Shortest path algorithm outline

The suggested procedure for identifying P independent paths in a network is outlined in the flow chart in Figure 2, and will be explained further in the following subsections. The method starts with loading the hydraulic model representation M of the network (see step 0. in Figure 2; the hydraulic model is assumed to contain:

- a set of $n_p$ links ($\mathbb{L}$), of which $\mathbb{P}$ is the subset of links which are pipes ($\mathbb{P} \subset \mathbb{L}$), and $\mathbb{B}$ is the subset of links which allow bi-directional flow ($\mathbb{B} \subset \mathbb{L}$) (as opposed to e.g. check valves). Each link $i$ has defined a first and second node ($\mathbf{N}_i^{(1)}$ and $\mathbf{N}_i^{(2)}$, respectively), a diameter ($D_i$), and a measure of its hydraulic resistance $f_i = f(k_i)$, where $k_i$ is the pipe's absolute roughness); pipes do in addition have a defined length ($L_i$).

- a set of nodes $\mathbb{N}$, of which $\mathbb{S}$ is the subset of nodes which are considered sources that can provide water into the system, i.e. reservoirs or water tanks ($\mathbb{S} \subset \mathbb{N}$)

Then each link $i$ in the network is assigned a weight $\widetilde{w_i}$ according to its hydraulic conductance (step 1):

- For the set of pipes ($\mathbb{P} \subset \mathbb{L}$), the weight is calculated to be proportional to the length ($L_i$) and some measure of the hydraulic resistance ($f_i$), and inversely proportional to some exponent of the diameter ($D_i^\gamma$) of the pipe. In this way, each pipe is assigned a weight according to its hydraulic resistance $\widetilde{w_i} = \frac{f_i L_i}{D_i^\gamma}$

- The weight for non-pipe links (valves and pumps; $\mathbb{L} \setminus \mathbb{P}$), is calculated in the same way, with the assumption that its length is twice its diameter, as is also assumed for open valves in EPANET (Rossman, 2000): $\widetilde{w_i} = \frac{2 f_i}{D_i^{\gamma-1}}$ .

Based on these weights, a weighted bi-directional graph, $G$, representation of the network is constructed (step 2.), in the following way:

- Each hydraulic node is represented as a graph vertex: $V \leftarrow \mathbb{N}$.

- All bi-directional links (i.e. all links which allow flow in both directions; $\mathbb{B} \subset \mathbb{L}$) are represented as two edges in the graph, one for each direction (one from $\mathbf{N}_i^{(1)}$ to $\mathbf{N}_i^{(2)}$ and one from $\mathbf{N}_i^{(2)}$ to $\mathbf{N}_i^{(1)}$ if $i \in \mathbb{B}$).

- All uni-directional links (i.e. all links which allow flow in only one direction; $\mathbb{L} \setminus \mathbb{B}$) are represented as one edge.

After the weighted graph has been constructed, one can start identifying paths for each node in the network. For each node $j$

which is not considered as a source ($\forall j \in \mathbb{N} \setminus \mathbb{S}$), the following steps are undertaken:

3. A copy of the graph is made: $\hat{G} \leftarrow G$.

4. For each path $p = 1,2 \dots P$, the shortest paths from all sources $\mathbb{S}$ to node $j$ are identified. The function $f_{SP}(\hat{G}, s, j)$ represents applying the Dijkstra's algorithm for finding the shortest path between node $j$ and $s$ in the graph $\hat{G}$. The function $f_{SP}$ returns $\mathbb{N}^{(s,j)}$, $\mathbb{L}^{(s,j)}$, and $W^{(s,j)}$, which is the set of nodes, set of links and the sum of

weights (total distance) in the path between $s$ and $j$, respectively.

5. Then, the shortest of the paths between $\mathbb{S}$ and $j$ is chosen as the $p$-shortest path: $\{\mathbb{N}_p^{(j)}, \mathbb{L}_p^{(j)}\} \leftarrow \{\mathbb{N}^{(s_p,j)}, \mathbb{L}^{(s_p,j)}\}$.

6. If the number of paths to be identified for node $j$ has not been reached ($p < P$), all the weights $\widehat{w_i}$ of the links that are in the current shortest path (going in both directions, thus $\mathbb{I} \cap \mathbb{L}_p^{(j)}$) are changed to the value $r_i$, where $r_i > \frac{L_i}{\min(L)} \sum_{\forall k \in \mathbb{L}} \widetilde{w_k}$, thus ensuring that a path going through any one of the elements in $\mathbb{L}_j^{(p)}$ will have a higher

weight than any non-looped path that does not go through any of the elements, and thereby prompting the shortest path algorithm to minimise the number of elements it has in common with the elements in $\mathbb{L}_j^{(p)}$. The algorithm then returns to identify the next $(p + 1)$ shortest path for node $j$. If the number of paths to been reached, the algorithm moves to the next $(j + 1)$ node.

Thus, by identifying the shortest path between two nodes, changing the weights of the edges in this path to a value that is

larger than the longest possible path in the graph ($r > \sum_{\forall i \in \mathbb{L}_p} \widetilde{w_i}$), then running the algorithm to find the shortest path again with these changed weights, the shortest path algorithm will identify a path that is as independent as possible from the paths that have already been identified.

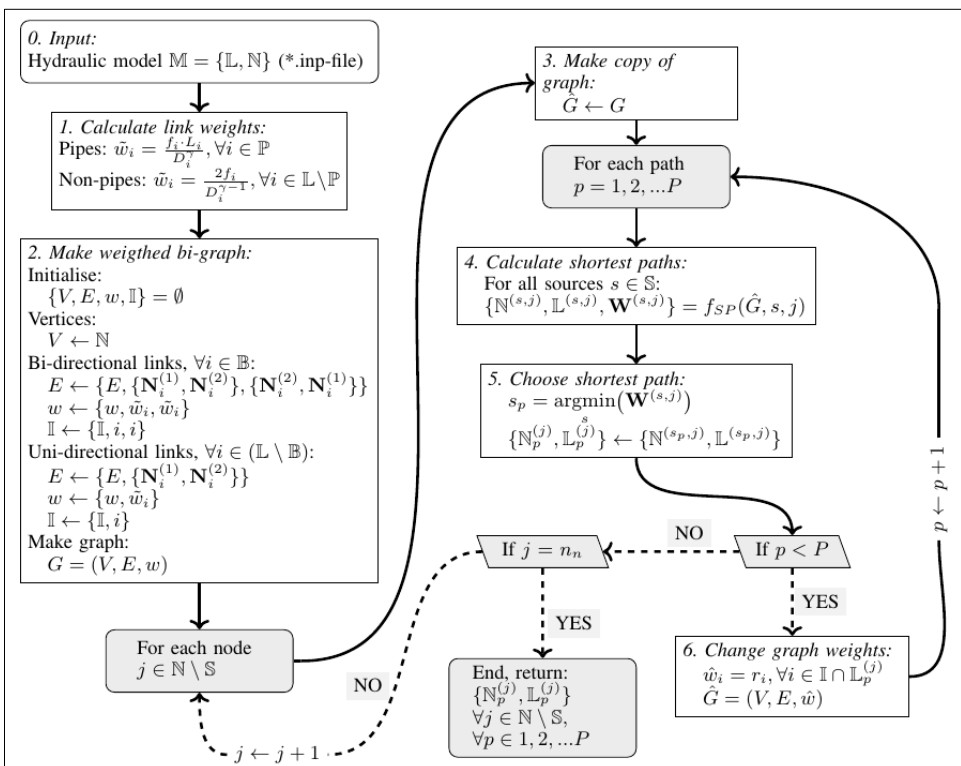

**Figure 2. Flow chart with suggested algorithm for identifying principal paths**

### 2.2.2 Performance metrics for independent paths

When the $P$ independent paths for a set of nodes in a network have been identified, one can utilise these paths to assess aspects of topological reliability and resilience in the network either for individual nodes, or on a sub-system or system level. When $P$ paths have been identified for any given node $j$, one can assess the degree of independence the paths to this node have from each other, by comparing which elements the $p$-th path has to its preceding paths. This can be calculated as follows, through the $p$-path independence proportion $I_{p,j}$:

$$I_{p,j}^{(\tilde{L})} = 1 - \frac{\sum_{\forall i \in \left(L_p^{(j)} \cap \{L_1^{(j)} \cup L_2^{(j)} \cup ... \cup L_{p-1}^{(j)}\}\right)} \tilde{L}_i}{\sum_{\forall i \in L_p^{(j)}} \tilde{L}_i}, \forall p > 1, \tag{1}$$

where $\tilde{L}_i$ is some measure of distance for link $i$, and may for instance pipe length $L_i$ or the probability of failure for each link $i$. As shown in Eq. (1), this index calculates the total length of elements in path $\mathbb{L}_p^{(j)}$ that are also present in any of the preceding paths ($\mathbb{L}_1^{(j)} \cup \mathbb{L}_2^{(j)} \cup ... \cup \mathbb{L}_{p-1}^{(j)}$) (i.e. elements that are not independent), and divides it with the total length of the path.

Hence, a value $I_{p,j} = 1$ means that the $p$-th path to node $j$ share none of the links from any of the preceding paths, and is therefore a supply to node $j$ which is completely independent of any of the other paths. Conversely, if $I_{p,j} = 0$, it means that

all of the links in the $p$-th path to $j$ are already present in one (or more) of the preceding paths, and that path $p$ does not provide any supply redundancy to $j$.

The independence of the $p$-th path can also be expressed in absolute terms:

$$A_{p,j}^{(\tilde{L})} = \sum_{\substack{\forall i \in (\mathbb{L}_p^{(j)} \cap \{ \mathbb{L}_1^{(j)} \cup \\ \mathbb{L}_2^{(j)} \cup \dots \cup \mathbb{L}_{p-1}^{(j)} \})}} \tilde{L}_i, \forall p > 1, \tag{2}$$

where $A_{p,j}^{(\tilde{L})}$ is the length the $p$-shortest path to node $j$ shares with a preceding path.

### 2.2.3 Simple network example

To illustrate how the suggested method in Figure 2 works, a fictitious example network is used (Figure 3), and the $P$-shortest paths between a single source $s$ and one node in this network is illustrated, identifying only the 3 shortest paths. This example network has pipe diameters according to Figure 3, and all pipes are assumed to have the same length ($L = 1000$ m) and friction factor ($f = 1$). The steps undertaken to identify the $P$ paths between two nodes ($s$ and $j$) are as follows:

- Step 1-2: Based on the network properties (pipe diameters and lengths), the weight of each edge is calculated and
the weighted bi-graph is constructed, as illustrated in Figure 4. (Although a bi-graph is constructed, only one link between each node pair is visible in the example figures, for simplicity of illustration.)

- Step 3: When starting the process of identifying the shortest paths to a new node $j$ a copy of the weighted graph is made ($\hat{G} \leftarrow G$), and

- Step 4-5: The shortest path between $s$ and $j$ is identified using the shortest path algorithm, yielding a result as
shown in Figure 5.

- Step 6: The weights of the links that are in this path are updated, and given a high weight, as illustrated in Figure 6, and the algorithm returns to step 4:

- Step 4(2): The shortest path algorithm is used again, with the altered weights, identifying the shortest path between $s$ and $j$, avoiding the links that have already been identified in previous paths, yielding a result as shown in Figure
205 7.

- If further paths to node $j$ are to be identified, the weights are updated again (see Figure 8), and the shortest path algorithm is used again to find the next shortest path (Figure 9). This process is continued until the $P$-shortest paths have been identified.

Table 1 shows the results from the analysis of the three shortest paths from $s$ to $j$. The indicators $I_{p,j}^{(L)}$ and $A_{p,j}^{(L)}$ shows that the
algorithm has been able to identify two paths which are completely independent of each other (path 1 and 2), since $I_{2,j}^{(L)} = 1.00$, and one path that is partially independent from the two preceding paths (path 3; $I_{2,j}^{(L)} = 0.80 < 1.00$).

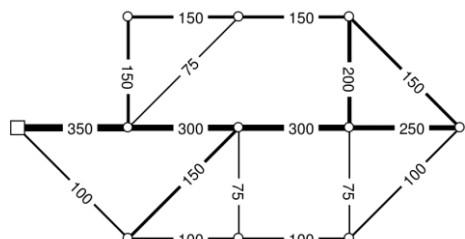

**Figure 3. Fictious example network, with pipe diameter labels**

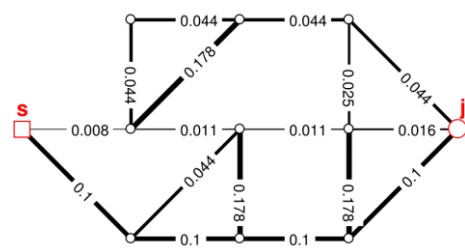

**Figure 4. Network represented as weighted graph with edge weights $w_i = L_i/D_i^2$**

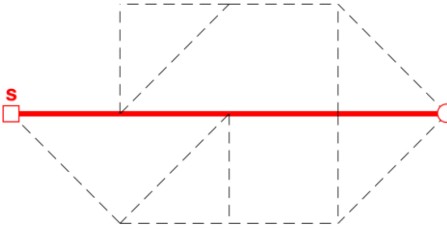

**Figure 5. First shortest path from $s$ to $j$ (solid line shows links in path, while dashed lines are not in path**

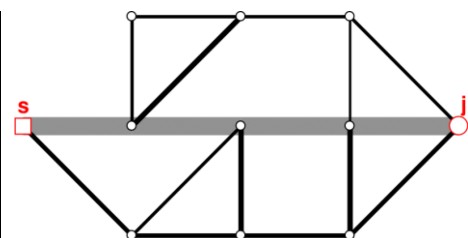

**Figure 6. Graph with updated weights after first shortest path is identified (thick grey lines indicate edges with updated weights**

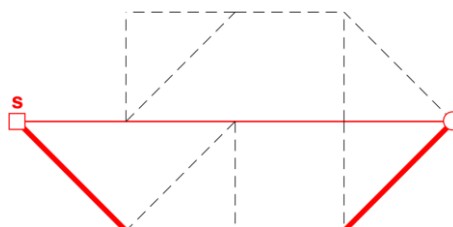

**Figure 7. First and second shortest paths from $s$ to $j$ (illustrated with thin and thick solid lines, respectively)**

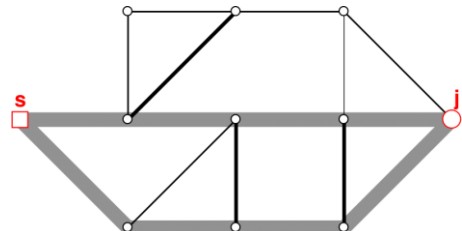

**Figure 8. Graph with updated weights after first and second shortest path are identified (thick gray lines indicate edges with updated weights)**

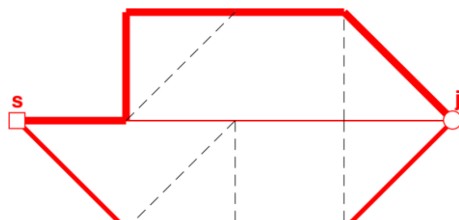

**Figure 9. First, second and third shortest paths from $s$ to $j$ (sorted from thinnest to thickest solid lines, respectively)**


**Table 1: Summary of results for the 3 shortest path analysis from $s$ to $j$, Figure 4.**

| Path ($p$) | $\sum_{\forall i \in L_p} w_i$ | $\sum_{\forall i \in L_p} L_i$ | $I_{p,j}^{(L)}$ | $A_{p,j}^{(L)}$ |
|---|---|---|---|---|
| 1 | 0.0464 | 4000 | - | - |
| 2 | 0.4000 | 4000 | 1.00 | 0 |
| 3 | 0.1859 | 5000 | 0.80 | 1000 |

## 2.3 Max flow algorithm-based variator that guides towards nearby minimal costs

The objective of the technique proposed here is to enhance the basic EA approach for sectorization with an additional variator. When applied to a given solution during, this variator picks one DMA from the solution and searches for nodes in its vicinity that may be optimal to include in it, based on the local network topology. On its own, the variator creates a greedy searching behavior around a solution, accepting the macroscopic structure of the DMAs and specifically aiming to reduce the number of boundaries between them. It is assumed that by carefully managing the rate at which this hybrid variator is applied to parts of individual solutions, solutions can be seeded with optimal substructures, while the other, basic variators provide enough variation to avoid early convergence to local optima.

### 2.3.1 Algorithm outline and simple network example

The individual steps of the variator are outlined below and illustrated in Figure 10. The lettered list matches the labels in the figure:

A. The algorithm choses a specific DMA (light grey node cluster) and takes not of its boundaries (red links) with the rest of the network (white nodes).
B. Every non-DMA node within a certain distance of the t DMA is found (drak grey nodes). The links that connect two nodes from this selection are collected as well. The subgraph that is formed by these nodes and links is used in the next steps.
C. The isolated subgraph is expanded with the links that originally connected the subgraph to the DMA (blue lines), which are instead connected to a virtual source node. Moreover, the isolated subgraph is expanded with the links that originally connected the subgraph to the rest of the network (red lines), which are instead connected to a virtual source node.
D. The Edmonds-Karp max flow algorithm (Dinic, 1970) is used to find the maximum number of independent paths (green lines) from the virtual source to the virtual sink through the isolated subgraph.
E. The nodes in the isolated subgraph that can be reached from the virtual source without using any link in the independent paths found in step D are found with a breadth first search from the source (blue nodes).
F. In the original DMA configuration, the nodes found in step E are assimilated into the original DMA.

In effect, the algorithm searches for local imperfections on the interface between the DMA and the rest of the network.

Firstly, boundaries that lead to the same 'chokepoint' a few nodes away can be reduced by including the nodes up to the

chokepoint (such as is the case for the assimilated node cluster on the right in the example). Achieving the same through random splitting and merging of the DMA and its neighbouring DMAs would take many generations in the EA.

Secondly, tiny DMAs that fall within the isolated subgraph entirely are joined to the DMA (such as the assimilated node cluster on the left in the example). This deliberately eradicates small node clusters from the solution, which would eventually occur through random merging otherwise.

Combined, these effects optimize the number of boundaries with minimal changes to the original DMA. The exact extend of the changes allowed can be controlled by choosing the depth of the subgraph around the DMA, which can be used as a parameter of the optimization algorithm.

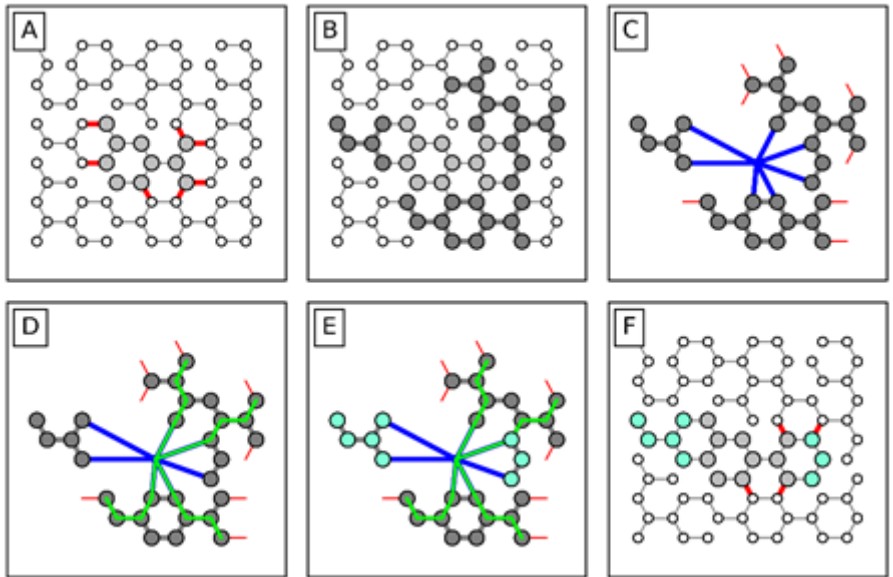

**Figure 10. Conceptual steps made by the hybrid variator to reduce the number of boundaries of a specific DMA by assimilation**
**certain nearby nodes. The individual steps are explained in more detail in the text.**

### 2.3.2 Case study

The performance of this approach was tested within the context of a case study involving the optimization of a real DWDS: the network of the city of The Hague, in the supply area of the Dutch water utility Dunea. At the moment of writing, The Hague's network is strongly meshed and has no DMAs implemented, other than one pilot DMA that separates ~2000

customers from the other ~48000. Dunea seeks to implement a DMA structure in The Hague as a part of their effort to better monitor the flow of their water supply.

The subdivision of this network into DMA's was originally optimized with a basic NSGA-II EA for two objectives: minimizing the total number of DMA boundaries and minimizing the maximum DMA size (in terms of daily peak demand).

Further details about the background and definition of the original optimization problem were previously described in (van Laarhoven and Gardien, 2019).

Here, an optimization over a limited number of generations (100) was repeated for different combinations of rates of occurrence assigned to the variators (mutation through merging/splitting of a single DMA, crossover by combining DMAs from two solutions and application of the hybrid variator described above to a single DMA). In the first tests (EA1 to EA9), only mutation and crossover were applied to roughly scan for the most advantageous basic settings. Then, for the most advantageous combination of basic mutation and crossover, additional tests (HGA1 to HGA4) were performed with varying rates of the hybrid variator included. The performed tests are summarized in Table 2.

**Table 2: algorithm settings during the experiments performed.**

| General algorithm settings | | | |
|---|---|---|---|
| Population | 100 | | |
| Elitism | 20% | | |
| Generations | 100 | | |
| Experiments | | | |
| Name | Merge/split rate | Crossover rate | Hybrid variator rate |
| EA1 | 0.03 | 0.1 | - |
| EA2 | 0.03 | 0.3 | - |
| EA3 | 0.03 | 0.9 | - |
| EA4 | 0.09 | 0.1 | - |
| EA5 | 0.09 | 0.3 | - |
| EA6 | 0.09 | 0.9 | - |
| EA7 | 0.27 | 0.1 | - |
| EA8 | 0.27 | 0.3 | - |
| EA9 | 0.27 | 0.9 | - |
| HGA1 | 0.27 | 0.1 | 0.03 |
| HGA2 | 0.27 | 0.1 | 0.09 |
| HGA3 | 0.27 | 0.1 | 0.27 |
| HGA4 | 0.27 | 0.1 | 0.81 |

The results of each test were abstracted in terms of the quality of the Pareto front (hypervolume (Cao et al., 2015) with respect to the network's combined base demand and the maximum number of boundaries between DMAs allowed by the optimization: 4413 $m^3/h$ and 500 boundaries, respectively. This allows to investigate the influence of the hybrid variator on searching behavior. The computational time of each test was registered to gain insight in the computational cost associated with the use of the hybrid variator. Each test was performed in triplo to account for reproducibility.

**3 Results**

**3.1 Result shortest path analyses**

An example result from the shortest independent path analyses performed on Trondheim's DWDS is demonstrated in Figure 11. In this example, the three shortest independent paths from available sources to one example node, $j$, have been identified. Node $j$ is situated centrally in a small pressure zone in the periphery of Trondheim's DWDS. The analysis shows that the

hydraulically closest source supplying node $j$ is the reservoir situated in the west of the pressure zone; the path from this source to node $j$ is indicated with a thick red line. Although there are multiple paths from the local reservoir to node $j$ which are shorter than the 2nd shortest path identified (cyan line), the 2nd shortest path is the only path which is completely independent of the 1st shortest path identified, as there are no paths which are completely independent of each other from the reservoir to node $j$. After the 1st and 2nd shortest independent paths have been identified, the algorithm finds that there are

no additional completely independent paths to node $j$ remaining, and therefore seeks to find the path which shares the minimum path length with the previously identified paths. The 3rd shortest path (green line) shares a subset of its path with the 1st shortest identified, but the algorithm seeks to find the solution where the 1st and 3rd path has the minimum length of elements in common. The results derived from the independent path search can be used as a constraint when optimizing the allocation of flow-altering valves in a DWDS.

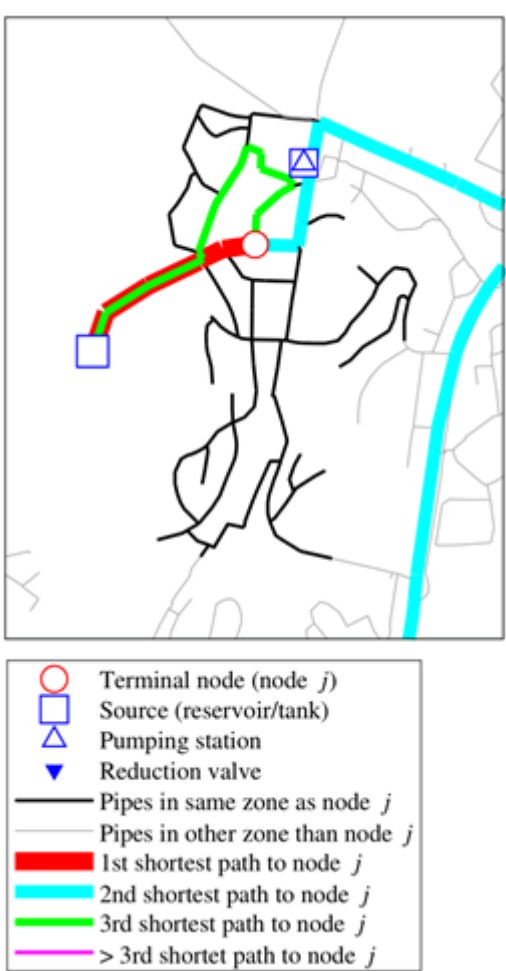


**Figure 11. Example of the three shortest paths to a node in Trondheim's DWDS**

The results obtained for the independent paths in a DWDS can be compared to the more traditional maxflow value measure. When the maxflow between all sources and a node in DWDS is calculated for an unweighted graph, the resulting maxflow value is equal to the number of completely independent paths between the sources and the node in question. Summarized for
Trondheim's DWDS, Figure 12 demonstrates the relation between the MF value between sources and each node in the system, and the length of the path limiting each node from achieving a higher max flow value. The system contains roughly 4700 nodes which only have one completely independent path from a source (red area in Figure 12). However, as the black line demonstrates, there are less than 2000 of these nodes where the 1st and 2nd shortest path shares more than 1 km of the path. Conversely, one can see from Figure 12 that there are a few hundred nodes, for which the single flow paths (where the
1st and 2nd shortest paths are shared) are longer than 5 km. Thus, the results in Figure 12 demonstrate how the proposed independent path measures can be used as a supplement to metrics describing one- or multisided supply in DWDSs.

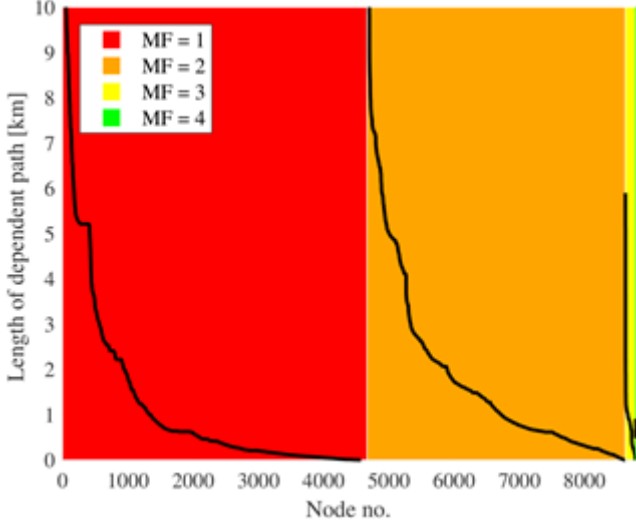

**Figure 12. Relation between MF and the length of path limiting the MF value for each node in Trondheim's DWDS**

### 3.2 Result of the max-flow variator case study

Figure 13 summarizes the combined results of the experiments. The performances of all solutions in the Pareto fronts of all experiments are plotted, grouped only by colour in terms of whether the hybrid variator was used or not. From this, it becomes apparent that, indeed, the algorithm is able to produce a set of solutions of significantly higher quality in the same number of iterations when the hybrid variator is used. At smaller DMA sizes, the number of DMA boundaries is reduced by 50-100. Especially from a practical point of view, this is a substantial increase in quality if one considers that the realization
of only a single DMA boundary may already cost a water utility thousands of euros.

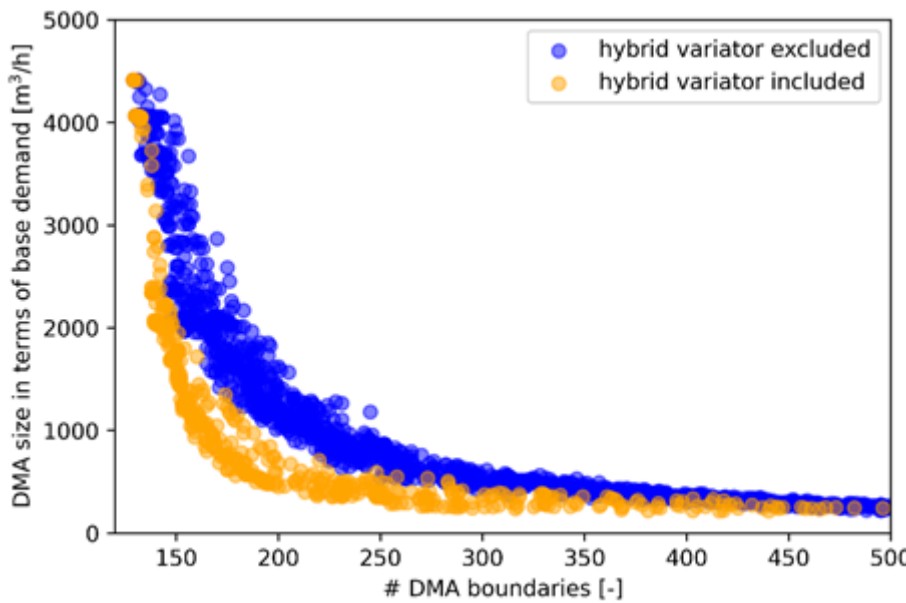

**Figure 13. Solutions of the Pareto fronts that followed from the experiments summarized in Table 2. Blue corresponds to the results from EA1 to EA9; orange corresponds to the results from HGA1 to HGA4.**

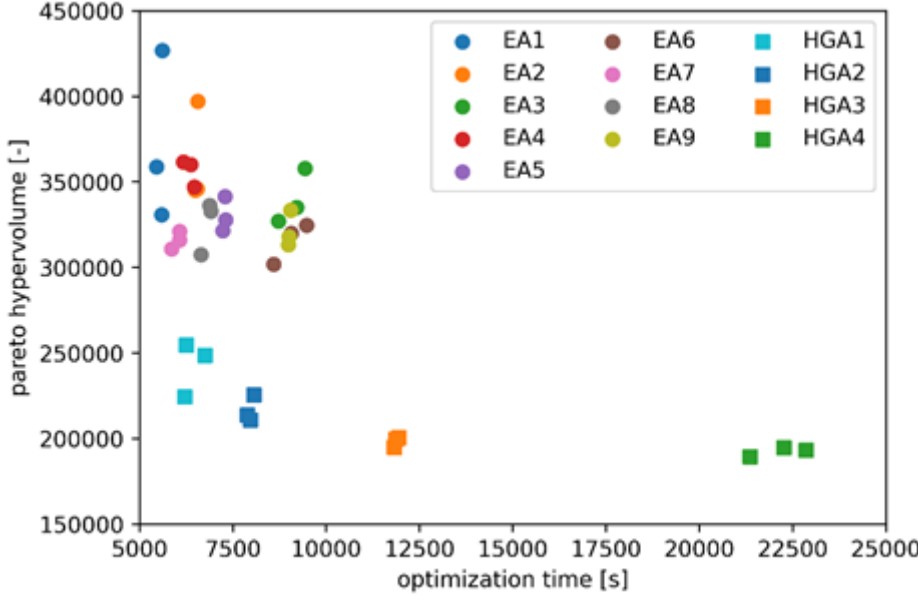

**Figure 14. The results of the experiments, summarized as the hypervolumes produced plotted against the computation time (each experiment was performed in triplo).**

The hypervolumes of the individual Pareto fronts are plotted against the computation times that were required to produce them in Figure 14. Experiment EA7 produced the lowest average Pareto volume while also having one of the lowest

computational times of the basic experiments. Hence, the settings for the merge, split and crossover rates of EA7 were used in the HGAx experiments as well.

The data demonstrates that using the hybrid variator with a rate of occurrence of only a few percent already leads to a substantial decrease in hypervolume (increase in quality). The quality can be increased with higher rates of occurrence, but this comes at a significant cost of computational time: HGA4 takes around 5 times longer than HGA1.

## 4 Conclusions

For the past decade, graph theory has become an increasingly popular tool for analysing and optimizing the performance of DWDSs. Among other things, graph theory has been used as a computationally efficient means for optimally sectorizing DWDSs into DMAs or other subnetworks. In most of the cases, graph theory-derived concepts – such as modularity indices, meshedness, centrality, algebraic connectivity, nodal degree, network density and so on – have been used as metrics to assess the efficiency and resilience of sectorization solutions, either during the optimization process or during a pre-processing step in which assumed optimal 'building blocks' of DMAs are identified (Bui et al. 2020). These approaches rely on a common assumption: that these theoretical, graph theory-based heuristics are in some way congruous with practical DWDS performance and therefore beneficial as a driver for optimization. The techniques described in in this chapter were designed to exploit graph theory concepts without the need for such an assumption, keeping the objectives and constraints explicitly defined in terms of the strategic goals of the water utility expert: a minimum number of required interventions (boundaries) while hydraulic function is ensured.

The results presented in this paper represent computationally quick ways of solving sectorization problems, while at the same time considering specific practical constraints. The shortest path algorithm presented in section 2.2 can be used as a pre-processing step that ultimately excludes pipes as viable locations for pressure control zone boundaries, with practical requirements and regulations in mind. This provides an approach to use EA for optimizing the design of pressure reduction zones - while guaranteeing acceptable performance under a multitude of possible failure scenarios - in a way that is computationally feasible.

The hybrid variator presented in section 2.3 can be used (sparingly) in addition to other variators to add a local search component to the search behaviour that contributes to finding stronger solutions more quickly with EA (as shown in figure 13). The variator can be used more rigorously to find even stronger solutions at the cost of substantial computational time (as shown in figure 14). As a result, the variator can be a valuable additional asset when applying EA in the water utility practice to optimize the design of DMAs.

The shortest path algorithm presented in section 2.2 essentially constitutes a search space reduction with a specific DMA functionality in mind and the hybrid variator presented in section 2.3 essentially constitutes a greedy optimization step towards a specific DMA property. Although the initial results look promising and are able to provide results that are suitable for the water utility practice, both approaches increase the risk for the optimization to get stuck in local optima (i.e. to arrive

at solutions that are 'very good' but not 'the best achievable'). Future work should focus on further elucidating this potential trade-off.

**Author contributions**

M.M.R. designed the approach described in section 2.2 and executed the case study described in section 3.1. K.vL designed the approach described in section 2.3 and executed the case study described in 3.2. M.M.R. and K.vL prepared the manuscript.

**Acknowledgements**

The authors thank Dennis Gardien, Ina Vertommen and Peter van Thienen for their roles in the Dutch case study, and Odd Atle Tveit, Birgitte G. Johannessen and Rannveig Høseggen for their roles in the Norwegian case study. Part of this research was funded by the BTO joined research programme of the Dutch water utilities. The research was also partly funded through the Universitetskommune collaboration between The Norwegian University of Science and Technology and Trondheim municipality.

**Conflicts of interest**

The authors declare no conflicts of interest.

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
