# Peer review of "Technical note: Graph theory-based heuristics to aid in the implementation of optimized drinking water network sectorization"

_Drinking Water Engineering and Science, 2021_

## Author Response (AR1)

*Dear Referees,*

*Thank you for taking the time to read our manuscript and thank you for your thorough and constructive comments. We appreciate the effort you have put into this and believe your comments will contribute to improve the quality of our manuscript. Please find answers to your comments and questions below. Your comments are written in black text, while our responses are provided using blue text.*

**RC1:**

The DWDS sectorization problem is not well defined, this should be corrected by adding an appropriate problem statement. The same applies to the optimization problem formulation. Instead, the authors go straight into the presentation of the solution method based on EA and Graph Theory, which makes things hard to follow.

We propose to expand on the problem statement in the introduction by adding the following to the statements in alinea 3 of the introduction: *"The key challenge of network sectorization lies in finding ways to efficiently divide the network is as many DMAs as possible with as few changes (which are costly) to the network as possible. This essentially is a version of the (np-hard) minimal k-cut problem (Kim et al. 2011). "* and then to also reemphasize the problem statement in the lead of the bullet list in 2.1 that descibes the layout of the optimization approach: " *The white boxes in Figure 1 illustrate a basic way in which an EA can be applied to find solutions to the sectorization problem, i.e. to find ways to divide the network into subnetworks with as few boundaries between them as possible:* "

Literature review is missing important recent publications on the topic of DWDS sectorization. This is not the first paper on this topic nor the one that makes use of Graph Theory. Recent Vasilic et al (2020) paper can be used as a good starting point for improved literature review as it contains relevant references. Authors are encouraged to use the improved review to better position their approach within the existing body of literature. This will also help better justify the novelty of the proposed method.

This is a valid point. We will expand on the literature in the introduction, by including references to literature both pertaining to the different motivations for pressure management and novel techniques for optimization of sectorization solutions. Some preliminary suggested literature to include may be:

[revised manuscript text omitted]

There is no discussion section in the technical note. Yes, the space available is limited but it would be good to, briefly, discuss the pros and cons, especially the limitations of proposed sectorization methodology.

One possible drawback of the approaches is added to the conclusions section, in relation to future work (see response to the final comment). In addition to mentioning this potential drawback, we also suggest adding a concluding remark about the strengths of what has been actually achieved, namely "computationally quick ways of solving sectorization problems, while at the same time considering specific practical constraints" in the conclusion.

The methodology proposed is not compared to any of the existing sectorization methods. This should be ideally done to support various claims made (see e.g. two claims made in the last paragraph of the Conclusions section). Otherwise, these claims need to be toned down.

We agree. As a comparison to other methods is not our intention, we tone down the claims to better match our goal: to provide additional approaches rather than replace other ones: " *The shortest path Algorithm presented in section 2.2 can be used as a pre-processing step that ultimately excludes pipes as viable locations for pressure control zone boundaries, with practical requirements and regulations in mind. This provides an approach to use EA for optimizing the design of pressure reduction zones - while guaranteeing acceptable performance under a multitude of possible failure scenarios - in a way that is computationally feasible. The hybrid variator presented in section 2.3 can be used (sparingly) in addition to other variators to add a local search component to the search behaviour that contributes to finding stronger solutions more quickly with EA (as shown in figure 13). The variator can be used more rigorously to find even stronger solutions at the cost of substantial computational time (as shown in figure 14). As a result, the variator can be a valuable additional asset when applying EA in the water utility practice to optimize the design of DMAs.* "

The conclusions section should also mention some future work.

We have added the following section to the conclusions: " *The shortest path algorithm presented in section 2.2 essentially constitutes a search space reduction with a specific DMA functionality in mind and the hybrid variator presented in section 2.3 essentially constitutes a greedy optimization step towards a specific DMA property. Although the initial results look promising and are able to provide results that are suitable for the water utility practice, both approaches increase the risk for the optimization to get stuck in local optima (i.e. to arrive at solutions that are 'very good' but not 'the best achievable'). Future work should focus on further elucidating this potential trade-off.* "

**RC2:**

The paper is well written, the figures explain very well the text, and the conclusions are solid. Well done.

Thank you

Some minor remarks.

Line 31: add "from" between "process" and "start"

Line 114: Is "Dijkstra's algorithm" so generally known that no reference is required?

Line 142: start numbering at 1.

We have addressed and changed the revised manuscript according to the abovementioned minor remarks.

---

## Author Response (AR2)

Dear Editor,

Thank you for your quick and positive processing of our revised manuscript. We have changed the typo ("is" to "in") in the sentence "The key challenge of network sectorization lies in finding ways to efficiently divide the network in as many DMAs as possible", according to your comment, in the revised manuscript uploaded together with this letter.

Best wishes,

Marius Rokstad & Karel van Laarhoven